# DENOISING DIFFUSION GAMMA MODELS

## ABSTRACT

Generative diffusion processes are an emerging and effective tool for image and speech generation. In the existing methods, the underlying noise distribution of the diffusion process is Gaussian noise. However, fitting distributions with more degrees of freedom could improve the performance of such generative models. In this work, we investigate other types of noise distribution for the diffusion process. Specifically, we introduce the Denoising Diffusion Gamma Model (DDGM) and show that noise from Gamma distribution provides improved results for image and speech generation. Our approach preserves the ability to efficiently sample state in the training diffusion process while using Gamma noise.

## 1 INTRODUCTION

Deep generative neural networks have shown significant progress over the last years. The main architectures for generation are: (i) VAE (Kingma & Welling, 2013) based, for example, NVAE (Vahdat & Kautz, 2020) and VQ-VAE (Razavi et al., 2019), (ii) GAN (Goodfellow et al., 2014) based, for example, StyleGAN (Karras et al., 2020) for vision application and WaveGAN (Donahue et al., 2018) for speech (iii) Flow-based, for example Glow (Kingma & Dhariwal, 2018) (iv) Autoregressive, for example, Wavenet for speech (Oord et al., 2016) and (v) Diffusion Probabilistic Models (Sohl-Dickstein et al., 2015), for example, Denoising Diffusion Probabilistic Models (DDPM) (Ho et al., 2020) and its implicit version DDIM (Song et al., 2020a).

Models from this last family have shown significant progress in generation capabilities in the last years, e.g., (Chen et al., 2020; Kong et al., 2020b), and have achieved results comparable to state-of-the-art generation architecture for both images and speech.

A DDPM is a Markov chain of latent variables. Two processes are modeled: (i) a diffusion process and (ii) a denoising process. During training, the diffusion process learns to transform data samples into Gaussian noise. Denoising is the reverse process and it is used during inference for generating data samples, starting from Gaussian noise. The second process can be conditioned on attributes to control the generation sample. To obtain high-quality synthesis, a large number of denoising steps is used (i.e. 1000 steps). A notable property of the diffusion process is a closed-form formulation of the noise that arises from accumulating diffusion stems. This allows sampling arbitrary states in the Markov chain of the diffusion process without calculating the previous steps.

In the Gaussian case, this property stems from the fact that adding Gaussian random variables leads to another Gaussian random variable. Other distributions have similar properties. For example, for the Gamma distribution, the sum of two random variables that share the scale parameter is a Gamma random variable of the same scale. The Poisson distribution has a similar property. However, its discrete nature makes it less suitable for DDPM.

In DDPM, the mean of the Gaussian random variables is set at zero. The Gamma random variable, with its two parameters (shape and scale), is better suited to fit the data than a Gaussian random variable with one degree of freedom (scale). Furthermore, the Gamma random variable generalizes other distributions, and many other distributions can be derived from it (Leemis & McQueston, 2008).

The added modeling capacity of the Gamma random variable can help speed up the convergence of the DDPM model. Consider, for example, a conventional DDPM model that was trained with Gaussian noise on the CelebA dataset (Liu et al., 2015).

The noise distribution throughout the diffusion process can be visualized by computing the histogram of the estimated residual noise in the generation process. The estimated residual noise $\hat{\epsilon}$ is given by

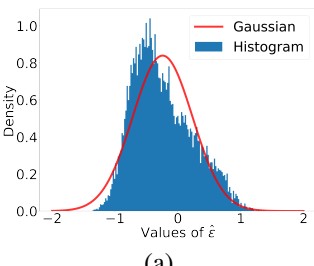 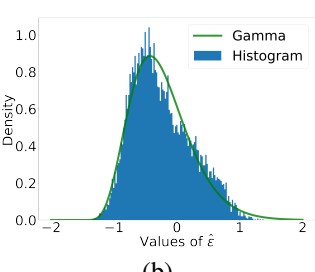 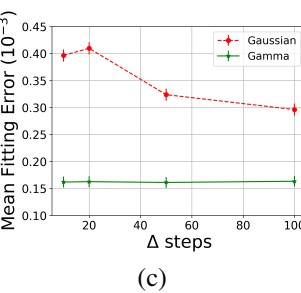

(a)              (b)              (c)

Figure 1: Fitting a distribution to the histogram of the generation error, which given by the scaled difference between $x_0$ and the image $x_t$ after $t$ DDPM steps $\hat{\epsilon} = \frac{\sqrt{\bar{\alpha}_t}x_0 - x_t}{\sqrt{1-|\bar{\alpha}_t|}}$. The model is a pretrained DDPM (Gaussian) celebA (64x64) model. (a) The fitting of a Gaussian to the histogram of a typical image after $t - 50$ steps. (b) Fitting a Gamma distribution. (c) The fitting error to Gaussian and Gamma distribution, measured as the MSE between the histogram and the fitted probability distribution function. Each point is the average value for the generation of 100 images. The vertical error bars denote the standard deviation.

$\hat{\epsilon} = \frac{\sqrt{\bar{\alpha}_t}x_0 - x_t}{\sqrt{1-|\bar{\alpha}_t|}}$, where $\bar{\alpha}_t$ is the noise schedule, $x_0$ is the data point and $x_t$ is the estimate state at timestep $t$, as can be derived from Eq.4 from (Song et al., 2020a). Both a Gaussian distribution and Gamma distribution can then be fitted to this histogram, as shown in Fig. 1(a,b). As can be seen, the Gamma distribution provides a better fit to the estimated residual noise $\hat{\epsilon}$. Moreover, Fig. 1(c) presents the mean fitting error between the histogram and the fitted probability distribution function. Evidently, the Gamma distribution is a better fit than the Gaussian distribution. While the model was trained to estimate Gaussian noise, at inference time it has to address a different distribution.

In this paper, we investigate the non-Gaussian Gamma noise distribution. As noted, this distribution seems to fit the histogram of the generation error better than the Gaussian distribution, and it also has favorable properties such as its behavior under addition and scalar multiplication. The proposed models maintain the property of the diffusion process of sampling arbitrary states without calculating the previous steps. Our results are demonstrated in two major domains: vision and audio. In the first domain, the proposed method is shown to provide a better FID score for generated images. For speech data, we show that the proposed method improves various measures, such as Perceptual Evaluation of Speech Quality (PESQ) and short-time objective intelligibility (STOI).

## 2 RELATED WORK

In their seminal work, Sohl-Dickstein et al. (2015) introduce the Diffusion Probabilistic Model. This model is applied to various domains, such as time series and images. The main drawback in the proposed model is that it needs up to thousands of iterative steps to generate a valid data sample. Song & Ermon (2019) proposed a diffusion generative model based on Langevin dynamics and the score matching method (Hyvärinen & Dayan, 2005). The model estimates the Stein score function (Liu et al., 2016) which is the gradient of the logarithm of data density. Given the Stein score function, the model can generate data points.

Denoising Diffusion Probabilistic Models (DDPM) (Ho et al., 2020) combine generative models based on score matching and neural Diffusion Probabilistic Models into a single model. Similarly, in Chen et al. (2020); Kong et al. (2020a) a generative neural diffusion process based on score matching was applied to speech generation. These models achieve state-of-the-art results for speech generation, and show superior results over well-established methods, such as Wavernn (Kalchbrenner et al., 2018), Wavenet (Oord et al., 2016), and GAN-TTS (Bińkowski et al., 2019).

Diffusion Implicit Models (DDIM) offer a way to accelerate the denoising process (Song et al., 2020a). The model employs a non-Markovian diffusion process to generate a higher quality sample. The model helps reduce the number of diffusion steps, e.g., from a thousand steps to a few hundred.

**Algorithm 1** DDPM training procedure.

1: Input: dataset $d$, diffusion process length $T$, noise schedule $\beta_1, ..., \beta_T$
2: **repeat**
3:     $x_0 \sim d(x_0)$
4:     $t \sim \mathcal{U}(\{1, ..., T\})$
5:     $\varepsilon \sim \mathcal{N}(0, I)$
6:     $x_t = \sqrt{\bar{\alpha}_t} x_0 + \sqrt{1 - \bar{\alpha}_t}\varepsilon$
7:     Take gradient descent step on:
      $\|\varepsilon - \varepsilon_\theta(x_t, t)\|_1$
8: **until** converged

**Algorithm 2** DDPM sampling algorithm

1: $x_T \sim \mathcal{N}(0, I)$
2: **for** t= T, ..., 1 **do**
3:     $z \sim \mathcal{N}(0, I)$
4:     $\hat{\varepsilon} = \varepsilon_\theta(x_t, t)$
5:     $x_{t-1} = \frac{x_t - \frac{1-\alpha_t}{\sqrt{1-\bar{\alpha}_t}}\hat{\varepsilon}}{\sqrt{\alpha_t}}$
6:     **if** $t \neq 1$ **then**
7:       $x_{t-1} = x_{t-1} + \sigma_t z$
8:     **end if**
9: **end for**
10: **return** $x_0$

Dhariwal & Nichol (2021) find a better diffusion architecture through a series of exploratory experiments, leading to the Ablated Diffusion Model (ADM). This model outperforms the state-of-the-art in image synthesis, which was previously provided by GAN based-models, such as BigGAN-deep (Brock et al., 2018) and StyleGAN2 (Karras et al., 2020). ADM is further improved using a novel Cascaded Diffusion Model (CDM). Our contribution is fundamental and can be incorporated into the proposed ADM and CDM architectures.

Watson et al. (2021) proposed an efficient method for sampling from diffusion probabilistic models by a dynamic programming algorithm that finds the optimal discrete time schedules. Choi et al. (2021) introduces the Iterative Latent Variable Refinement (ILVR) method for guiding the generative process in DDPM. Moreover, Kong & Ping (2021) systematically investigates fast sampling methods for diffusion denoising models. Lam et al. (2021) propose bilateral denoising diffusion models (BDDM), which take significantly fewer steps to generate high-quality samples.

Huang et al. (2021) derive a variational framework for likelihood estimating of the marginal likelihood of continuous-time diffusion models. Moreover, Kingma et al. (2021) shows equivalence between various diffusion processes by using a simplification of the variational lower bound (VLB).

Song et al. (2020b) show that score-based generative models can be considered a solution to a stochastic differential equation. Gao et al. (2020) provide an alternative approach for training an energy-based generative model using a diffusion process.

Another line of work in audio is that of neural vocoders based on a denoising diffusion process. WaveGrad (Chen et al., 2020) and DiffWave (Kong et al., 2020a) are conditioned on the mel-spectrogram and produce high-fidelity audio samples, using as few as six steps of the diffusion process. These models outperform adversarial non-autoregressive baselines. Popov et al. (2021) propose a text-to-speech diffusion base model, which allows generating speech with the flexibility of controlling the trade-off between sound quality and inference speed.

Diffusion models were also applied to natural language processing tasks. Hoogeboom et al. (2021) proposed a multinomial diffusion process for categorical data and applied it to language modeling. Austin et al. (2021) generalize the multinomial diffusion process with Discrete Denoising Diffusion Probabilistic Models (D3PMs) and improve the generated results for the text8 and One Billion Word (LM1B) datasets.

## 3 DIFFUSION MODELS FOR GAMMA DISTRIBUTION

We start by recapitulating the Gaussian case, after which we derive diffusion models for the Gamma distribution.

### 3.1 BACKGROUND - GAUSSIAN DDPM

Diffusion networks learn the gradients of the data log density:

$$s(y) = \nabla_y \log p(y) \tag{1}$$

By using Langevin Dynamics and the gradients of the data log density $\nabla_y \log p(y)$, a sample procedure from the probability can be done by:

$$\tilde{y}_{i+1} = \tilde{y}_i + \frac{\eta}{2} s(\tilde{y}_i) + \sqrt{\eta} z_i \tag{2}$$

where $z_i \sim \mathcal{N}(0, I)$ and $\eta > 0$ is the step size.

The diffusion process in DDPM (Ho et al., 2020) is defined by a Markov chain that gradually adds Gaussian noise to the data according to a noise schedule. The diffusion process is defined by:

$$q(x_{1:T}|x_0) = \prod_{t=1}^{T} q(x_t|x_{t-1}), \tag{3}$$

where T is the length of the diffusion process, and $x_T, ..., x_t, x_{t-1}, ..., x_0$ is a sequence of latent variables with the same size as the clean sample $x_0$. The Diffusion process is parameterized with a set of parameters called noise schedule $(\beta_1, \ldots \beta_T)$, which defines the variance of the noise added at each step:

$$q(x_t|x_{t-1}) := \mathcal{N}(x_t; \sqrt{1 - \beta_t} x_{t-1}, \beta_t \mathbf{I}), \tag{4}$$

Since we are using a Gaussian noise random variable at each step, the diffusion process can be simulated for any number of steps with the closed formula:

$$x_t = \sqrt{\bar{\alpha}_t} x_0 + \sqrt{1 - \bar{\alpha}_t} \varepsilon, \tag{5}$$

where $\alpha_i = 1 - \beta_i$, $\bar{\alpha}_t = \prod_{i=1}^{t} \alpha_i$ and $\varepsilon = \mathcal{N}(0, \mathbf{I})$.

Diffusion models are a class of generative neural network of the form $p_\theta(x_0) = \int p_\theta(x_{0:T}) dx_{0:T}$ that learn to reverse the diffusion process. One can write that:

$$p_\theta(x_{0:T}) = p(x_T) \prod_{t=1}^{T} p_\theta(x_{t-1}|x_t) \tag{6}$$

As described in (Ho et al., 2020), one can learn to predict the noise present in the data with a network $\varepsilon_\theta$ and sample from $p_\theta(x_{t-1}|x_t)$ using the following formula:

$$x_{t-1} = \frac{x_t - \frac{1-\alpha_t}{\sqrt{1-\bar{\alpha}_t}} \varepsilon_\theta(x_t, t)}{\sqrt{\bar{\alpha}_t}} + \sigma_t \varepsilon, \tag{7}$$

where $\varepsilon$ is white noise and $\sigma_t$ is the standard deviation of added noise. (Song et al., 2020a) use $\sigma_t^2 = \beta_t$.

The training procedure of $\varepsilon_\theta$ is defined in Alg.1. Given the input dataset $d$, the algorithm samples $\epsilon$, $x_0$ and $t$. The noisy latent state $x_t$ is calculated and fed to the DDPM neural network $\varepsilon_\theta$. A gradient descent step is taken in order to estimate the $\varepsilon$ noise with the DDPM network $\varepsilon_\theta$. The objective for the diffusion model is a variational bound on the model data log likelihood.

The complete inference algorithm present at Alg. 2. Starting from Gaussian noise and then reversing the diffusion process step-by-step, by iteratively employing the update rule of Eq. 7. To perform generation with few denoising iterations one can use the update equation introduced in Song et al. (2020a). This work greatly improves the results of diffusion networks when performing sampling with few generative steps.

$$x_{n-1} = \sqrt{\bar{\alpha}_{n-1}} \hat{x}_{0,n} + \sqrt{1 - \bar{\alpha}_{n-1} - \tilde{\sigma}_n^2} \varepsilon_\theta(x_n, \bar{\alpha}_n) + \tilde{\sigma}_n \varepsilon, \tag{8}$$

Intuitively this equation changes the added noise for the generative steps. It uses a blend of the noise from the previous state ($\varepsilon_\theta(x_n, \bar{\alpha}_n)$) and random noise ($\varepsilon$). One can write $\tilde{\sigma} = \eta \sqrt{\beta_n (1 - \bar{\alpha}_{n-1})(1 - \bar{\alpha}_n)}$ this allows to have a simple parameter $\eta$ to choose the ratio of the blend. On top of improving the results in short samplings regimes, this allows to generate in a deterministic way when using $\eta = 0$.

### 3.2 DENOISING DIFFUSION GAMMA MODELS (DDGM)

We expand the framework of diffusion generative processes by incorporating a new noise distribution, namely the Gamma Distribution. We call this new type of models Denoising Diffusion Gamma Models. First, we define the Gamma diffusion process, then we present a way to sample from this process, and finally we show how to train those models by computing the variational lower bound and deriving a novel loss function from it.

#### 3.2.1 THE GAMMA MODEL

In the Gaussian case the diffusion equation (Eq. 4) can be written as:

$$x_t = \sqrt{1 - \beta_t} x_{t-1} + \sqrt{\beta_t} \epsilon_t \tag{9}$$

where $\epsilon_t$ is the Gaussian noise of step $t$. One can denote $\Gamma(k, \theta)$ as the Gamma distribution, where $k$ and $\theta$ are the shape and the scale respectively. We modify Eq. 9 by adding, during the diffusion process, noise that follows a Gamma distribution:

$$x_t = \sqrt{1 - \beta_t} x_{t-1} + (g_t - \mathbb{E}(g_t)) \tag{10}$$

where $g_t \sim \Gamma(k_t, \theta_t)$, $\theta_t = \sqrt{\bar{\alpha}_t}\theta_0$ and $k_t = \dfrac{\beta_t}{\alpha_t \theta_0^2}$. Note that $\theta_0$ and $\beta_t$ are hyperparameters (and the noise term has zero mean.)

Since the sum of Gamma random variables (with the same scale parameter) is distributed as Gamma distribution, one can derive a closed form for $x_t$, i.e. an equation to calculate $x_t$ from $x_0$:

$$x_t = \sqrt{\bar{\alpha}_t} x_0 + (\bar{g}_t - \bar{k}_t \theta_t) \tag{11}$$

where $\bar{g}_t \sim \Gamma(\bar{k}_t, \theta_t)$ and $\bar{k}_t = \sum_{i=1}^{t} k_i$.

**Lemma 1.** *Let $\theta_0 \in \mathbb{R}$, Assuming $\forall t \in \{1, ..., T\}$, $k_t = \dfrac{\beta_t}{\alpha_t \theta_0^2}$, $\theta_t = \sqrt{\bar{\alpha}_t}\theta_0$, and $g_t \sim \Gamma(k_t, \theta_t)$. Then $\forall t \in \{1, ..., T\}$ the following hold:*

$$E(g_t - E(g_t)) = 0, V(g_t - E(g_t)) = \beta_t \tag{12}$$

$$x_t = \sqrt{\bar{\alpha}_t} x_0 + (\bar{g}_t - E(\bar{g}_t)) \tag{13}$$

*where $\bar{g}_t \sim \Gamma(\bar{k}_t, \theta_t)$ and $\bar{k}_t = \sum_{i=1}^{t} k_i$*

The complete proof for Lemma 1 is given in Appendix A.1.

Similarly to Eq.7, the inference is given by:

$$x_{t-1} = \frac{x_t - \frac{1-\alpha_t}{\sqrt{1-\bar{\alpha}_t}} \varepsilon_\theta(x_t, t)}{\sqrt{\bar{\alpha}_t}} + \sigma_t \frac{\bar{g}_t - E(\bar{g}_t)}{\sqrt{V(\bar{g}_t)}} \tag{14}$$

In Algorithm 3 we describe the training procedure. As input we have the: (i) initial scale $\theta_0$, (ii) the dataset $d$, (iii) the maximum number of steps in the diffusion process $T$ and (iv) the noise schedule $\beta_1, ..., \beta_T$. The training algorithm sample: (i) an example $x_0$, (ii) number of step $t$ and (iii) noise $\varepsilon$. Then it calculates $x_t$ from $x_0$ by using Eq.11. The neural network $\varepsilon_\theta$ has an input $x_t$ and is conditional on the time step $t$. Next, it takes a gradient descent step to approximate the normalized noise $\frac{\bar{g}_t - \bar{k}_t \theta_t}{\sqrt{1-|\bar{\alpha}_t|}}$ with the neural network $\varepsilon_\theta$. The main changes between Algorithm 3 and the single Gaussian case (i.e. Alg. 1) are the following: (i) calculating the Gamma parameters, (ii) $x_t$ update equation and (iii) the gradient update equation.

The inference procedure is given in Algorithm 4. It starts from a zero mean noise $x_T$ sampled from $\Gamma(\theta_T, \bar{k}_T)$. Next, for $T$ steps the algorithm estimates $x_{t-1}$ from $x_t$ by using Eq.14. Note that as in (Song et al., 2020a) $\sigma_t = \beta_t$. Algorithm 4 replaces the Gaussian version (i.e. Alg. 2) with the following: (i) the starting sampling point $x_T$, (ii) the sampling noise $z$ and (iii) the $x_t$ update equation.

**Algorithm 3** Gamma Training Algorithm

1: Input: initial scale $\theta_0$, dataset $d$, diffusion process length $T$, noise schedule $\beta_1, ..., \beta_T$
2: **repeat**
3:  $\quad x_0 \sim d(x_0)$
4:  $\quad t \sim \mathcal{U}(\{1, ..., T\})$
5:  $\quad \bar{g}_t \sim \Gamma(\bar{k}_t, \theta_t)$
6:  $\quad x_t = \sqrt{\bar{\alpha}_t} x_0 + (\bar{g}_t - \bar{k}_t \theta_t)$
7:  $\quad$ Take a gradient descent step on:
$$\left| \frac{\bar{g}_t - \bar{k}_t \theta_t}{\sqrt{1 - |\bar{\alpha}_t|}} - \varepsilon_\theta(x_t, t) \right|$$
8: **until** converged

**Algorithm 4** Gamma Inference Algorithm

1: $\gamma \sim \Gamma(\theta_T, \bar{k}_T)$
2: $x_T = \gamma - \theta_T * \bar{k}_T$
3: **for** t = T, ..., 1 **do**
4:  $\quad x_{t-1} = \frac{x_t - \frac{1 - \alpha_t}{\sqrt{1 - \bar{\alpha}_t}} \epsilon(x_t, t)}{\sqrt{\alpha_t}}$
5:  $\quad$ **if** t > 1 **then**
6:  $\quad\quad z \sim \Gamma(\theta_{t-1}, \bar{k}_{t-1})$
7:  $\quad\quad z = \frac{z - \theta_{t-1} \bar{k}_{t-1}}{\sqrt{(1 - \bar{\alpha}_t)}}$
8:  $\quad\quad x_{t-1} = x_{t-1} + \sigma_t z$
9:  $\quad$ **end if**
10: **end for**

### 3.2.2 THE REVERSE PROCESS FOR DDGM

The reverse process $q(x_{t-1}|x_0, x_t)$ defines the underlying generation process. Therefore, in this section, we will obtain the reverse process for the Gamma denoising diffusion model. Furthermore, we will use the reverse process $q(x_{t-1}|x_0, x_t)$ to obtain the variational lower bound and the appropriate loss function for the Gamma distribution denoising diffusion model.

**Lemma 2.** *Denote $q(x_{t-1}|x_0, x_t)$ as the reverse process of the proposed Gamma diffusion model. Then, the reverse process is proportional to:*

$$q(x_{t-1}|x_0, x_t) \propto \frac{X_t^{k_t - 1} e^{-X_t/\theta_t} \bar{X}_{t-1}^{\bar{k}_{t-1} - 1} e^{-\bar{X}_{t-1}/\theta_{t-1}}}{\bar{X}_t^{\bar{k}_t - 1} e^{-\bar{X}_t/\theta_t}} \tag{15}$$

*where*

*1. $X_t = x_t - \sqrt{1 - \beta_t} x_{t-1} + k_t \theta_t$*

*2. $\bar{X}_t = x_t - \sqrt{\bar{\alpha}_t} x_0 + \bar{k}_t \theta_t$*

*3. $\bar{X}_{t-1} = x_{t-1} - \sqrt{\bar{\alpha}_{t-1}} x_0 + \bar{k}_{t-1} \theta_{t-1}$*

The complete proof for Lemma 2 is given in Appendix A.1. It states that the reverse process is proportional to three Gamma random variables $X_t$, $\bar{X}_{t-1}$, and $\bar{X}_t$. This observation allows us to develop the associated variational lower bound.

### 3.2.3 VARIATIONAL LOWER BOUND FOR DDGM

Denoising diffusion models (Ho et al., 2020) trained by optimizing the usual variational bound on negative log likelihood:

$$E\left[-log(p_\theta(x_0)\right] \leq E_q\left[-\log p(x_T) - \sum_{t \geq 1} \log \frac{p_\theta(x_{t-1}|x_t)}{q(x_t|x_{t-1})}\right] = L_{VLB} \tag{16}$$

To get the variational lower bound for the proposed Gamma denoising diffusion model, one can use Eq.5 from Ho et al. (2020):

$$L_{VLB} = E_q\left[L_T + \sum_{t > 1} L_{t-1} + L_0\right] \tag{17}$$

where $L_T, L_{t-1}$ and $L_0$ define by:

1. $L_T = D_{KL}(q(x_T|x_0)||q(x_T))$

2. $L_{t-1} = D_{KL}(q(x_{t-1}|x_0, x_t)||p_\theta(x_{t-1}|\hat{x}_0, x_t))$

3. $L_0 = -\log(p_\theta(x_0|x_1))$

It should be noted that in the Gaussian case the KL terms have a closed form. $L_T$ is constant and ignored during training since it doesn't have learnable parameters. Moreover, in (Ho et al., 2020) $L_0$ modeled with discrete decoder, however, in our proposed model we empirically found that the impact $L_0$ is negligible and can be removed.

Therefore, to calculate the variatonal lower bound one needs to obtain:

$$L_{t-1} = D_{KL}(q(x_{t-1}|x_0, x_t)||p_\theta(x_{t-1}|\hat{x}_0, x_t)) \tag{18}$$

where:

$$\hat{x}_0 = \frac{x_t - \sqrt{1 - \bar{\alpha}_t}\varepsilon_\theta(x_t, t)}{\sqrt{\bar{\alpha}_t}} \tag{19}$$

**Lemma 3.** *The $L_{t-1}$ for the proposed Gamma diffusion model is upper bounded by the following $L_1$ norm:*

$$L_{t-1} \leq \left(C_1 + C_2 + \frac{C_3}{\bar{g}_t} + \frac{C_4}{\bar{g}_{t-1}}\right)|x_0 - \hat{x}_0| \tag{20}$$

*where $C_1, C_2, C_3$ and $C_4$ are constant terms.*

The complete proof for Lemma 3 is given in Appendix A.1.

As can be seen, the variational lower bound is bounded by some constant forms multiplied by the L1 norm between the data point $x_0$ and its estimation $\hat{x}_0$. The constant terms $C_1, C_2, C_3$ and $C_4$ as well as $\bar{g}_t$ and $\bar{g}_{t-1}$ are known values during the training.

### 3.2.4 LOSS FUNCTION FOR DDGM

Denoising diffusion probabilistic models use the variational lower bound to minimize the negative log likelihood. As described in Sec.3.2.1, one can minimize the variational lower bound by $L_t$ for $t \geq 1$. To do so, one can minimize the L1 norm from Eq.34. Our model optimizes the L1 norm between the sampled noise $\epsilon_\theta$ and the estimated noise $\varepsilon_\theta$. This is verified in the following lemmas.

**Lemma 4.** *Minimizing the variational lower bound for DDGM (i.e. $L_t$ for $t \geq 1$) is equivalent to minimizing the L1 norm between the sampled noise and the estimated noise:*

$$\mathcal{L} = \left|\frac{\bar{g}_t - \bar{k}_t\theta_t}{\sqrt{1 - \bar{\alpha}_t}} - \varepsilon_\theta(x_t, t)\right| \tag{21}$$

The complete proof for Lemma 4 is given in Sec.A.4 at the appendix. Thus, the loss that is used in the Alg.3 is given by $\mathcal{L} = \left|\frac{\bar{g}_t - \bar{k}_t\theta_t}{\sqrt{1 - \bar{\alpha}_t}} - \varepsilon_\theta(x_t, t)\right|$.

## 4 EXPERIMENTS

### 4.1 SPEECH GENERATION

For our speech experiments we used a version of Wavegrad (Chen et al., 2020) based on this implementation Vovk (2020) (under BSD-3-Clause License). We evaluate our model with high-level perceptual quality of speech measurements, PESQ (Rix et al., 2001) and STOI (Taal et al., 2011). We used the standard Wavegrad method with the Gaussian diffusion process as a baseline. We use two Nvidia Volta V100 GPUs to train our models.

For all the experiments, the inference noise schedules ($\beta_0, .., \beta_T$) were defined as described in the Wavegrad paper (Chen et al., 2020). For 1000 and 100 iterations the noise schedule is linear, for 25 iterations it comes from the Fibonacci and for 6 iterations we performed a model-dependent grid search to find the best noise schedule parameters. For other hyper-parameters (e.g. learning rate, batch size, etc) we use the same as in Wavegrad (Chen et al., 2020). Training was performed using the following form of Eq. 10, e.g. $\theta_t = \sqrt{\bar{\alpha}_t}\theta_0$ and $k_t = \frac{\beta_t}{\bar{\alpha}_t\theta_0{}^2}$. Our best results were obtained using $\theta_0 = 0.001$.

Table 1: PESQ and STOI metrics for the LJ dataset for various Wavegrad-like models.

| Model \ Iteration | PESQ (↑) | | | | STOI (↑) | | | |
|---|---|---|---|---|---|---|---|---|
| | 6 | 25 | 100 | 1000 | 6 | 25 | 100 | 1000 |
| WaveGrad (Chen et al., 2020) | 2.78 | 3.194 | 3.211 | 3.290 | 0.924 | 0.957 | 0.958 | 0.959 |
| DDGM (ours) | **3.07** | **3.208** | **3.214** | **3.308** | **0.948** | **0.972** | **0.969** | **0.969** |

Table 2: FID (↓) score comparison for CelebA(64x64) dataset. Lower is better.

| Model \ Iteration | 10 | 20 | 50 | 100 | 1000 |
|---|---|---|---|---|---|
| DDPM (Ho et al., 2020) | 299.71 | 183.83 | 71.71 | 45.2 | **3.26** |
| DDGM - Gamma Distribution DDPM (ours) | **35.59** | **28.24** | **20.24** | **14.22** | 4.09 |
| DDIM (Song et al., 2020a) | 17.33 | 13.73 | 9.17 | 6.53 | 3.51 |
| DDGM - Gamma Distribution DDIM (ours) | **11.64** | **6.83** | **4.28** | **3.17** | **2.92** |

Table 3: FID (↓) score comparison for LSUN Church (256x256) dataset. Lower is better. The results of DDIM for $T = 1000$ are not reported by Song et al. (2020a).

| Model \ Iteration | 10 | 20 | 50 | 100 | 1000 |
|---|---|---|---|---|---|
| DDPM (Ho et al., 2020) | 51.56 | 23.37 | 11.16 | 8.27 | 7.89 |
| DDGM - Gamma Distribution DDPM (ours) | **28.56** | **19.68** | **10.53** | **7.87** | **6.91** |
| DDIM (Song et al., 2020a) | 19.45 | 12.47 | 10.84 | 10.58 | NA |
| DDGM - Gamma Distribution DDIM (ours) | **18.11** | **11.32** | **10.31** | **8.75** | **7.34** |

Table 4: FID (↓) score comparison for ImageNet (64x64) dataset. Lower is better.

| Model \ Iteration | 10 | 20 | 50 | 100 | 1000 |
|---|---|---|---|---|---|
| DDIM (Song et al., 2020a) | 42.88 | 35.40 | 31.98 | 30.74 | 28.81 |
| DDGM - Gamma Distribution DDIM (ours) | **42.17** | **31.84** | **28.75** | **27.02** | **24.22** |

Table 5: FID (↓) score comparison during the inference process for CelebA (64x64) dataset for the full inference procedure and for an earlier stage in the process.

| Iteration Model | T=10 (first 5 steps/10 steps) | T=20 (first 10 steps/20 steps) | T=100 (first 50 steps/100 steps) |
|---|---|---|---|
| DDIM (Song et al., 2020a) | 54.32/17.33 | 43.35/13.73 | 36.68/6.53 |
| DDGM (ours) | 55.40/11.64 | 42.43/6.83 | 35.20/3.17 |

**Results** Tab. 1 presents the PESQ and STOI measurement for the LJ dataset (Ito & Johnson, 2017). As can be seen, for the proposed Gamma denoising diffusion model our results are better than the Wavegrad baseline for all number of iterations in both PESQ and STOI.

## 4.2 IMAGE GENERATION

Our model is based on the DDIM implementation available in (Jiaming Song & Ermon, 2020) (under the MIT license). We trained our model on three image datasets (i) CelebA 64x64 (Liu et al., 2015), (ii) LSUN Church 256x256 (Yu et al., 2015) and (iii) ImageNet 64x64 (Deng et al., 2009). The Fréchet Inception Distance (FID) (Heusel et al., 2017) is used as the benchmark metric. For all experiments, similarly to previous work (Song et al., 2020a), we compute the FID score with $50,000$ generated images, using the torch-fidelity implementation (Obukhov et al., 2020). Similar to (Song et al., 2020a), the training noise schedule $\beta_1, ..., \beta_T$ is linear with values raging from $0.0001$ to $0.02$. For other hyperparameters (e.g. learning rate, batch size etc) we use the same parameters that appear in DDPM (Ho et al., 2020). We use eight Nvidia Volta V100 GPUs to train our models. The $\theta_0$ parameter for Gamma distribution set to $0.001$.

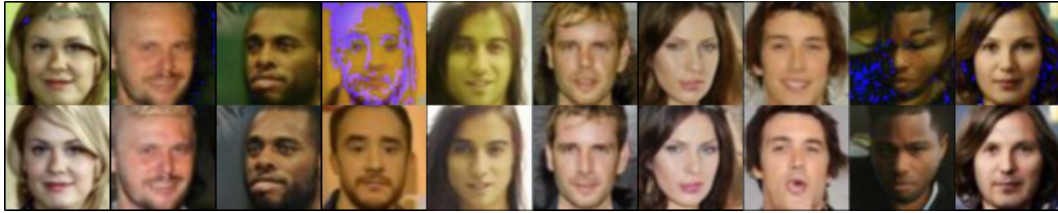

Figure 2: Typical examples of images generated with $100$ iterations and $\eta = 0$. For models trained with different noise distributions - (i) First row - Gaussian noise and (ii) Second row - Gamma noise. All models start from the same noise instance.

**Results** We test our models with the inference procedure from DDPM (Ho et al., 2020) and DDIM (Song et al., 2020a). In Tab. 2 we provide the FID score for CelebA (64x64) dataset (Liu et al., 2015) (under non-commercial research purposes license). As can be seen for DDPM inference procedure for $10, 20, 50, 100$ steps, the best results were obtained from the Gamma model, which improves results by a gap of $264$ FID scores for ten iterations. For $100$ iterations, the Gamma model improves results by 31 FID scores. For $1000$ iterations, the best results were obtained from the DDPM model. Nevertheless, our Gamma model obtains results that are closer to the DDPM by a gap of $0.83$. For the DDIM procedure, the best results were obtained with the Gamma model for all number of iterations. Fig. 2 presents samples generated by the three models. Our models provide better quality images when compared to DDPM and DDIM methods.

In Tab. 3 we provide the FID score for the LSUN church dataset (Yu et al., 2015). As can be seen, the Gamma model improves results over the baseline for $10, 20, 50, 100, 1000$ iterations.

Tab. 4 lists the FID score for the ImageNet 64x64 dataset. The Gamma model obtains better results than the DDIM baseline method for all iteration counts. Figure 3 compares between our proposed DDGM and the baseline DDIM method. The random samples are generated with $1000$ iterations with the DDIM generation algorithm and $\eta = 0$. In each of the nine instances, the same random noise is used as $x_{1000}$ for both models.

In Tab. 5, we present results for training the network on a Markov chain of size $T$ and perform all the processes until step $T/2$, from which we infer $x_0$ in one inference step. As can be seen, after the $T/2$ first iterations, there is no significant difference between Gaussian and Gamma networks. However, after the remaining $T/2$ iterations, the Gamma based model obtains a performance GAP over the Gaussian model. This suggests that our method is improving the results mostly during the final stage of the inference process.

## 5 CONCLUSIONS

We present a novel Gamma diffusion model. The model employs a Gamma noise distribution. A key enabler for using these distributions is a closed-form formulation (Eq. 11) of the multi-step noising process, which allows for efficient training. We also present the reverse process and the variational lower bound for the Gamma diffusion model. The proposed model improves the quality of generated image and audio, as well as the speed of generation in comparison to conventional, Gaussian-based diffusion processes. Our DDGM methods shows that diffusion model can benefit from non-Gaussian noise distributions. This comes at a cost of adding one new hyperparameter to tune ($\theta_0$). Working with other probability distributions, such as mixture models, may improve results even further.

## REPRODUCIBILITY STATEMENT

We provide in the supplementary file the complete code that was used to perform all of our experiments. This archive includes audio samples and the code for both image and speech experiments. Hyperparameters choices are clearly stated in Sec. 4 and the values are obtained from publicly available implementation of previous work. The proof of all the theoretical results are available in the appendix or are derived in the paper.

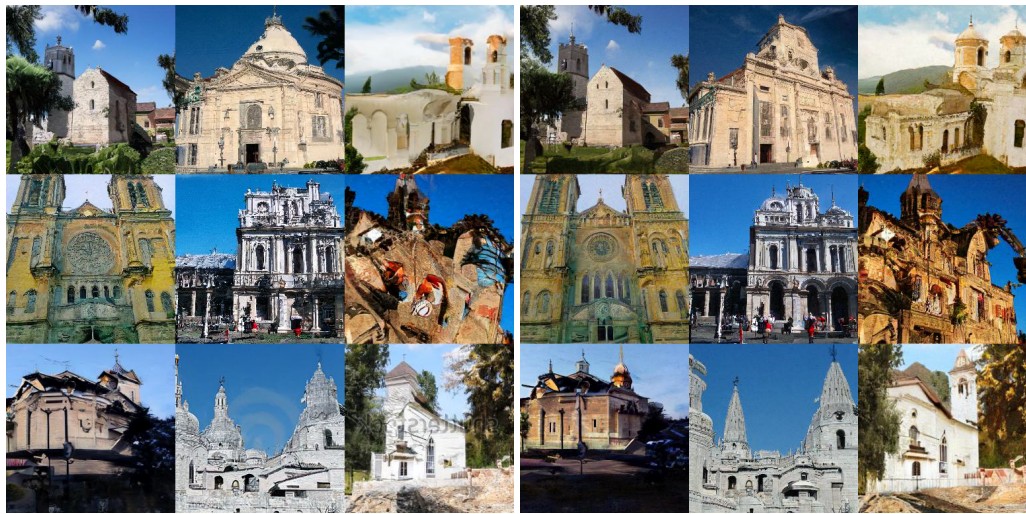

Figure 3: Comparison of generated samples with the (left) Gaussian noise (DDIM) and (right) Gamma noise (DDGM). In each of the nine cases, the two models start from the same noise instance, which leads to similar output images between the models.

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

# A    PROOFS

## A.1    PROOF OF LEMMA 1

**Lemma 1.** *Let $\theta_0 \in \mathbb{R}$, Assuming $\forall t \in \{1, ..., T\}$, $k_t = \dfrac{\beta_t}{\alpha_t \theta_0^2}$, $\theta_t = \sqrt{\bar{\alpha}_t}\theta_0$, and $g_t \sim \Gamma(k_t, \theta_t)$. Then $\forall t \in \{1, ..., T\}$ the following hold:*

$$E(g_t - E(g_t)) = 0, V(g_t - E(g_t)) = \beta_t \tag{12}$$

$$x_t = \sqrt{\bar{\alpha}_t}x_0 + (\bar{g}_t - E(\bar{g}_t)) \tag{13}$$

*where $\bar{g}_t \sim \Gamma(\bar{k}_t, \theta_t)$ and $\bar{k}_t = \sum_{i=1}^{t} k_i$*

*Proof.* The first part of Eq. 12 is immediate. The variance part is also straightforward:

$$V(g_t - E(g_t)) = k_t \theta_t^2 = \beta_t$$

Eq. 13 is proved by induction on $t \in \{1, ...T\}$. For $t = 1$:

$$x_1 = \sqrt{1 - \beta_1}x_0 + g_1 - E(g_1)$$

since $\bar{k}_1 = k_1$, $\bar{g}_1 = g_1$. We also have that $\sqrt{1 - \beta_1} = \sqrt{\bar{\alpha}_1}$. Thus we have:

$$x_1 = \sqrt{\bar{\alpha}_1}x_0 + (\bar{g}_1 - E(\bar{g}_1))$$

Assume Eq. 13 holds for some $t \in \{1, ...T\}$. The next iteration is obtained as

$$x_{t+1} = \sqrt{1 - \beta_{t+1}}x_t + g_{t+1} - E(g_{t+1}) \tag{22}$$
$$= \sqrt{1 - \beta_{t+1}}(\sqrt{\bar{\alpha}_t}x_0 + (\bar{g}_t - E(\bar{g}_t))) + g_{t+1} - E(g_{t+1}) \tag{23}$$
$$= \sqrt{\bar{\alpha}_{t+1}}x_0 + \sqrt{1 - \beta_{t+1}}\bar{g}_t + g_{t+1} - (\sqrt{1 - \beta_{t+1}}E(\bar{g}_t) + E(g_{t+1})) \tag{24}$$

It remains to be proven that (i) $\sqrt{1 - \beta_{t+1}}\bar{g}_t + g_{t+1} = \bar{g}_{t+1}$ and (ii) $\sqrt{1 - \beta_{t+1}}E(\bar{g}_t) + E(g_{t+1}) = E(\bar{g}_{t+1})$. Since $\bar{g}_t \sim \Gamma(\bar{k}_t, \theta_t)$ hold, then:

$$\sqrt{1 - \beta_{t+1}}\bar{g}_t \sim \Gamma(\bar{k}_t, \sqrt{1 - \beta_{t+1}}\theta_t) = \Gamma(\bar{k}_t, \theta_{t+1})$$

Therefore, we prove (i):

$$\sqrt{1 - \beta_{t+1}}\bar{g}_t + g_{t+1} \sim \Gamma(\bar{k}_t + k_{t+1}, \theta_{t+1}) = \Gamma(\bar{k}_{t+1}, \theta_{t+1})$$

which implies that $\sqrt{1 - \beta_{t+1}}\bar{g}_t + g_{t+1}$ and $\bar{g}_{t+1}$ have the same probability distribution.

Furthermore, by the linearity of the expectation, one can obtain (ii):

$$\sqrt{1 - \beta_{t+1}}E(\bar{g}_t) + E(g_{t+1}) = E(\sqrt{1 - \beta_{t+1}}\bar{g}_t + g_{t+1})$$
$$= E(\bar{g}_{t+1})$$

Thus, we have:

$$x_{t+1} = \sqrt{\bar{\alpha}_{t+1}}x_0 + (\bar{g}_{t+1} - E(\bar{g}_{t+1}))$$

which ends the proof by induction. □

## A.2  PROOF OF LEMMA 2

**Lemma 2.** *Denote $q(x_{t-1}|x_0, x_t)$ as the reverse process of the proposed Gamma diffusion model. Then, the reverse process is proportional to:*

$$q(x_{t-1}|x_0, x_t) \propto \frac{X_t^{k_t-1}e^{-X_t/\theta_t}\bar{X}_{t-1}^{\bar{k}_{t-1}-1}e^{-\bar{X}_{t-1}/\theta_{t-1}}}{\bar{X}_t^{\bar{k}_t-1}e^{-\bar{X}_t/\theta_t}} \tag{15}$$

*where*

1. $X_t = x_t - \sqrt{1 - \beta_t}x_{t-1} + k_t\theta_t$
2. $\bar{X}_t = x_t - \sqrt{\bar{\alpha}_t}x_0 + \bar{k}_t\theta_t$
3. $\bar{X}_{t-1} = x_{t-1} - \sqrt{\bar{\alpha}_{t-1}}x_0 + \bar{k}_{t-1}\theta_{t-1}$

*Proof.* The reverse process is given by:

$$q(x_{t-1}|x_0, x_t) = q(x_t|x_{t-1}, x_0)\frac{q(x_{t-1}|x_0)}{q(x_t|x_0)} \tag{25}$$

Next, one can calculate each one of the three main components of the reverse process, i.e. (i) $q(x_t|x_{t-1}, x_0)$, (ii) $q(x_{t-1}|x_0)$ and (iii) $q(x_t|x_0)$. Since $q$ is memoryless, $q(x_t|x_{t-1}, x_0) = q(x_t|x_{t-1})$. Therefore, the first component (i) of Eq. 25 is the forward process. The forward process is given by:

$$q(x_t|x_{t-1}) = p(g_t = x_t - \sqrt{1 - \beta_t}x_{t-1} + k_t\theta_t) \tag{26}$$

$$= \frac{(x_t - \sqrt{1 - \beta_t}x_{t-1} + k_t\theta_t)^{k_t - 1} e^{-(x_t - \sqrt{1 - \beta_t}x_{t-1} + k_t\theta_t)/\theta_t}}{\Gamma(k_t)\theta_t^{k_t}} \tag{27}$$

The second component of Eq.25 is given by:

$$q(x_{t-1}|x_0) = \frac{(x_{t-1} - \sqrt{\bar{\alpha}_{t-1}}x_0 + \bar{k}_{t-1}\theta_{t-1})^{\bar{k}_{t-1} - 1} e^{-(x_t - \sqrt{\bar{\alpha}_{t-1}}x_0 + \bar{k}_{t-1}\theta_{t-1})/\theta_{t-1}}}{\Gamma(\bar{k}_{t-1})\theta_t^{\bar{k}_{t-1}}} \tag{28}$$

Similarly, the third component of Eq.25 is given by:

$$q(x_t|x_0) = p(\bar{g}_t = x_t - \sqrt{\bar{\alpha}_t}x_0 + \bar{k}_t\theta_t) = \frac{(x_t - \sqrt{\bar{\alpha}_t}x_0 + \bar{k}_t\theta_t)^{\bar{k}_t - 1} e^{-(x_t - \sqrt{\bar{\alpha}_t}x_0 + \bar{k}_t\theta_t)/\theta_t}}{\Gamma(\bar{k}_t)\theta_t^{\bar{k}_t}} \tag{29}$$

Overall, the reverse process $q(x_{t-1}|x_0, x_t)$ is given by:

$$q(x_{t-1}|x_0, x_t) = \frac{\left((x_t - \sqrt{1 - \beta_t}x_{t-1} + k_t\theta_t)^{k_t - 1} e^{-(x_t - \sqrt{1 - \beta_t}x_{t-1} + k_t\theta_t)/\theta_t}\right)}{\Gamma(k_t)\theta_t}$$
$$\cdot \frac{\left((x_{t-1} - \sqrt{\bar{\alpha}_{t-1}}x_0 + \bar{k}_{t-1}\theta_{t-1})^{\bar{k}_{t-1} - 1} e^{-(x_t - \sqrt{\bar{\alpha}_{t-1}}x_0 + \bar{k}_{t-1}\theta_{t-1})/\theta_{t-1}}\right)}{\Gamma(\bar{k}_{t-1})\theta_t^{\bar{k}_{t-1}}} \tag{30}$$
$$\cdot \frac{\Gamma(\bar{k}_t)\theta_t^{\bar{k}_t}}{\left((x_t - \sqrt{\bar{\alpha}_t}x_0 + \bar{k}_t\theta_t)^{\bar{k}_t - 1} e^{-(x_t - \sqrt{\bar{\alpha}_t}x_0 + \bar{k}_t\theta_t)/\theta_t}\right)}$$

One can denote:

1. $X_t = x_t - \sqrt{1 - \beta_t}x_{t-1} + k_t\theta_t$

2. $\bar{X}_t = x_t - \sqrt{\bar{\alpha}_t}x_0 + \bar{k}_t\theta_t$

3. $\bar{X}_{t-1} = x_{t-1} - \sqrt{\bar{\alpha}_{t-1}}x_0 + \bar{k}_{t-1}\theta_{t-1}$

Thus, the reverse process $q(x_{t-1}|x_0, x_t)$ is proportional to:

$$q(x_{t-1}|x_0, x_t) \propto \frac{X_t^{k_t - 1} e^{-X_t/\theta_t} \bar{X}_{t-1}^{\bar{k}_{t-1} - 1} e^{-\bar{X}_{t-1}/\theta_{t-1}}}{\bar{X}_t^{\bar{k}_t - 1} e^{-\bar{X}_t/\theta_t}} \tag{31}$$

$\square$

### A.3 PROOF OF LEMMA 3

**Lemma 3.** *The $L_{t-1}$ for the proposed Gamma diffusion model is upper bounded by the following $L_1$ norm:*

$$L_{t-1} \leq \left(C_1 + C_2 + \frac{C_3}{\bar{g}_t} + \frac{C_4}{\bar{g}_{t-1}}\right)|x_0 - \hat{x}_0| \tag{20}$$

*where $C_1, C_2, C_3$ and $C_4$ are constant terms.*

*Proof.* We can calculate the $L_{t-1}$ with the exact form:

$$L_{t-1} = D_{KL}(q(x_{t-1}|x_0,x_t)||p_\theta(x_{t-1}|\hat{x}_0,x_t)) = E_{q(x_{t-1}|x_0,x_t)} \log\left(\frac{q(x_{t-1}|x_0,x_t)}{p_\theta(x_{t-1}|\hat{x}_0,x_t)}\right) \quad (32)$$

Using Eq.15 the RHS of Eq.32 become:

$$\log\left(\frac{q(x_{t-1}|x_0,x_t)}{p_\theta(x_{t-1}|\hat{x}_0,x_t)}\right) = (\bar{k}_{t-1}-1)\log(\frac{\bar{X}_{t-1}}{\hat{X}_{t-1}}) - \frac{\bar{X}_{t-1}-\hat{X}_{t-1}}{\theta_{t-1}} - (\bar{k}_t-1)\log(\frac{\bar{X}_t}{\hat{X}_t}) + \frac{\bar{X}_t-\hat{X}_t}{\theta_t}$$
$$(33)$$

One can show that the four terms present in the previous equation can be upper bounded with the L1 distance between the predicted $\hat{x}_0$ and the ground truth $x_0$:

- $|\frac{\bar{X}_{t-1}-\hat{X}_{t-1}}{\theta_{t-1}}| = |(x_0-\hat{x}_0)\frac{\sqrt{\bar{\alpha}_{t-1}}}{\theta_{t-1}}| \leq C_1|x_0-\hat{x}_0|$

- $|\frac{\bar{X}_t-\hat{X}_t}{\theta_t}| = |(x_0-\hat{x}_0)\frac{\sqrt{\bar{\alpha}_t}}{\theta_t}| \leq C_2|x_0-\hat{x}_0|$

- $(\bar{k}_t \quad - \quad 1)\log(\frac{\bar{X}_t}{\hat{X}_t}) \quad = \quad (\bar{k}_t \quad - \quad 1)\log\left(\frac{x_t-\sqrt{\bar{\alpha}_t}x_0+\bar{k}_t\theta_t}{x_t-\sqrt{\bar{\alpha}_t}\hat{x}_0+\bar{k}_t\theta_t}\right) \quad = $
$\log\left(1+\frac{\sqrt{\bar{\alpha}_t}(x_0-\hat{x}_0)}{x_t-\sqrt{\bar{\alpha}_t}\hat{x}_0+\bar{k}_t\theta_t}\right) \leq |\frac{\sqrt{\bar{\alpha}_t}(x_0-\hat{x}_0)}{x_t-\sqrt{\bar{\alpha}_t}\hat{x}_0+\bar{k}_t\theta_t}| = \frac{C_3}{\bar{g}_t}|x_0-\hat{x}_0|$

- $(\bar{k}_{t-1}-1)\log(\frac{\bar{X}_{t-1}}{\hat{X}_{t-1}}) = \log\left(1+\frac{\sqrt{\bar{\alpha}_{t-1}}(x_0-\hat{x}_0)}{x_{t-1}-\sqrt{\bar{\alpha}_{t-1}}\hat{x}_0+\bar{k}_{t-1}\theta_{t-1}}\right)$
$\leq |\frac{\sqrt{\bar{\alpha}_{t-1}}(x_0-\hat{x}_0)}{x_{t-1}-\sqrt{\bar{\alpha}_{t-1}}\hat{x}_0+\bar{k}_{t-1}\theta_{t-1}}| = \frac{C_4}{\bar{g}_{t-1}}|x_0-\hat{x}_0|$

The complete form of the $L_{t-1}$ upper bound can be expressed as follows:

$$L_{t-1} \leq E_{q(x_{t-1}|x_0,x_t)}\left(C_1+C_2+\frac{C_3}{\bar{g}_t}+\frac{C_4}{\bar{g}_{t-1}}\right)|x_0-\hat{x}_0| = \left(C_1+C_2+\frac{C_3}{\bar{g}_t}+\frac{C_4}{\bar{g}_{t-1}}\right)|x_0-\hat{x}_0|$$
$$(34)$$
$\square$

### A.4 PROOF OF LEMMA 4

**Lemma 4.** *Minimizing the variational lower bound for DDGM (i.e. $L_t$ for $t \geq 1$) is equivalent to minimizing the L1 norm between the sampled noise and the estimated noise:*

$$\mathcal{L} = \left|\frac{\bar{g}_t-\bar{k}_t\theta_t}{\sqrt{1-\bar{\alpha}_t}}-\varepsilon_\theta(x_t,t)\right| \quad (21)$$

*Proof.* From Eq.34, the variational lower bound of DDGM is given by $L_{t-1} \leq \left(C_1+C_2+\frac{C_3}{\bar{g}_t}+\frac{C_4}{\bar{g}_{t-1}}\right)|x_0-\hat{x}_0|$. Substitute Eq.19 and Eq.11 to the variational lower bound

we have:

$$L_{t-1} \leq \left( C_1 + C_2 + \frac{C_3}{\bar{g}_t} + \frac{C_4}{\bar{g}_{t-1}} \right) |x_0 - \hat{x}_0| \tag{35}$$

$$= \left( C_1 + C_2 + \frac{C_3}{\bar{g}_t} + \frac{C_4}{\bar{g}_{t-1}} \right) \left| x_0 - \frac{x_t - \sqrt{1 - \bar{\alpha}_t}\varepsilon_\theta(x_t, t)}{\sqrt{\bar{\alpha}_t}} \right| \tag{36}$$

$$= \left( C_1 + C_2 + \frac{C_3}{\bar{g}_t} + \frac{C_4}{\bar{g}_{t-1}} \right) \frac{1}{\sqrt{\bar{\alpha}_t}} \left| \sqrt{\bar{\alpha}_t}x_0 - x_t + \sqrt{1 - \bar{\alpha}_t}\varepsilon_\theta(x_t, t) \right| \tag{37}$$

$$= \left( C_1 + C_2 + \frac{C_3}{\bar{g}_t} + \frac{C_4}{\bar{g}_{t-1}} \right) \frac{1}{\sqrt{\bar{\alpha}_t}} \left| \sqrt{\bar{\alpha}_t}x_0 - \sqrt{\bar{\alpha}_t}x_0 - (\bar{g}_t - \bar{k}_t\theta_t) + \sqrt{1 - \bar{\alpha}_t}\varepsilon_\theta(x_t, t) \right| \tag{38}$$

$$= \left( C_1 + C_2 + \frac{C_3}{\bar{g}_t} + \frac{C_4}{\bar{g}_{t-1}} \right) \frac{1}{\sqrt{\bar{\alpha}_t}} \left| (\bar{g}_t - \bar{k}_t\theta_t) - \sqrt{1 - \bar{\alpha}_t}\varepsilon_\theta(x_t, t) \right| \tag{39}$$

$$= \left( C_1 + C_2 + \frac{C_3}{\bar{g}_t} + \frac{C_4}{\bar{g}_{t-1}} \right) \frac{\sqrt{1 - \bar{\alpha}_t}}{\sqrt{\bar{\alpha}_t}} \left| \frac{\bar{g}_t - \bar{k}_t\theta_t}{\sqrt{1 - \bar{\alpha}_t}} - \varepsilon_\theta(x_t, t) \right| \tag{40}$$

Since we are minimizing the variational lower bound, one can drop the constant term $\left( C_1 + C_2 + \frac{C_3}{\bar{g}_t} + \frac{C_4}{\bar{g}_{t-1}} \right) \frac{\sqrt{1 - \bar{\alpha}_t}}{\sqrt{\bar{\alpha}_t}}$. Therefore, minimizing the variational lower bound is equal to minimizing the term $\left| \frac{\bar{g}_t - \bar{k}_t\theta_t}{\sqrt{1 - \bar{\alpha}_t}} - \varepsilon_\theta(x_t, t) \right|$. $\qquad \square$

