# OpenReview forum: "Denoising Diffusion Gamma Models"
_ICLR.cc/2022/Conference — ICLR 2022 Submitted_

### Official Review · Reviewer_y4Qp · 2021-10-30

**Correctness:** 3
**Technical Novelty And Significance:** 3
**Empirical Novelty And Significance:** 2
**Recommendation:** 6
**Confidence:** 4

**Main Review:**

The derivation of the proposed DDGM model is sound, but the experiments are not sufficient enough.

1. Some obvious mistakes are needed to be corrected. For example, Eq. (6) and the denominator of Eq. (16).

2. Eq. (13) represents the parameterization for the reverse process $p_{\theta}(x_{t-1}|x_t)$. According to my understanding, all the terms of $\bar{g}_t$ in Eq. (13) should be replaced with $g_t$, please have a check.

3. The notations in Sec. 3.2.3 are confused. In Eq. (21), the reverse process is represented with $p_{\theta}(\cdot)$, but in the following part the reverse and the diffusion process are both denoted by $q(\cdot)$, which are very hard to read and understand. What's more, some notations in this section are not explained, such as the $\hat{x}_0$ in Eq.(24). In the related literatures of diffusion model, the diffusion and the reverse process are always denoted by $q(\cdot)$ and $p(\cdot)$ to distinguish. I suggest to reformulate this part following this common settings.

4. The experiments is weaken. Even though the final FID scores justify the superiorities of the proposed DDGM, I can't intuitively understand the necessity of introducing the Gamma assumption in the diffusion model. If you can give more visual result to prove such necessity, it will be better.



**Summary Of The Paper:**

Tho goal of this work is to replace the original Gaussian noise distribution in DDPM with Gamma distribution, since the Gamma distribution is with more degrees of freedom. To achieve this goal, it reformulates the diffusion and the corresponding reverse process, and also deduces the variational lower bound.

**Summary Of The Review:**

Even though the idea is interesting, this paper should be further improved from the mathematical formulations and the experiments. If so, I tend to increase my score.

---

> ### Author Response · Authors · 2021-11-18
> **Response to Reviewer y4Qp**
>
> Thank you for the very helpful suggestions for corrections and improvements. All of these are now implemented.
>
> Q: Some obvious mistakes are needed to be corrected. For example, Eq. (6) and the denominator of Eq. (16).
>
> A: Thank you for pointing this out, obviously, this is the probability of the sequence. We changed it to $p(x_{0:T})$ and it should be $\theta_t^{k_t}$.
>
> Q: Eq. (13) represents the parameterization for the reverse process $p_\theta(x_{t−1}|x_t)$. According to my understanding, all the terms of $\bar g_t$ in Eq. (13) should be replaced with $g_t$, please have a check.
>
> A: It is rightfully $\bar g_t$. The idea is that the added noise should match the distribution that the network was trained on. This is why it needs to be $\bar g_t$.
>
> Q: The notations in Sec. 3.2.3 are confused.
>
> A: Thank you. Following the review, we now use the p_theta for the reverse process and q for the forward process. We have fixed the notation in Sec 3.2.3. Regarding Eq.24, it is the definition of the notation for $\hat x_0$ that is used in Eq.23.
>
> Q: The experiments is weaken.
>
> A: In the paper, we show that Gaussian noise is not the best choice to use in diffusion models by both checking the histogram of noise for sample datasets and presenting empirical results using the Gamma distribution.
> Our main goal is not to suggest the Gamma distribution, but to open the door to non-Gaussian distributions. This includes, for example, choices such as mixture distribution. Unfortunately, one has to derive the exact Markovian processes to each such distribution.
> Regarding the experiments, at the request of reviewer C6QK, we have conducted experiments on ImageNet 64x64 which is a more diverse dataset. As the table below shows, our method improves by the same FID gap across multiple numbers of iterations. We add this table as Tab.4 in the revised manuscript.
>
> | Imagenet 64x64 (FID) | T=10  | T=20  | T=50  | T=100 | T=1000 |
> |----------------------|-------|-------|-------|-------|--------|
> | Gaussian (DDIM)      | 42.88 | 35.40 | 31.98 | 30.74 | 28.81  |
> | Gamma (DDIM)         | **42.17** | **31.84** | **28.75** | **27.02** | **24.22**  |

---

### Official Review · Reviewer_YK5T · 2021-10-31

**Correctness:** 3
**Technical Novelty And Significance:** 3
**Empirical Novelty And Significance:** 3
**Recommendation:** 5
**Confidence:** 3

**Main Review:**

Strengths:
  - Overall a sound idea with some strong experimental results.

Weakness:
  - Experimental results are not well presented and discussed. More insights from the experimental results are expected.

Comments:
  - Figure 2: Which model is used as the baseline, DDPM or DDIM? The samples from the baseline look significant worse than those from either Ho et al., 2020 or Song et al., 2020. Any idea?
  - Image experiments: since experiments on higher resolution were conducted (LSUN Church, 256x256), can you provide such image samples besides the lower resolution experiments?
  - Speech experiment: The audio samples with few iterations sounds significantly better than the baseline. However, all the audio samples from the proposed model with any number of iterations contain similar white-noise-like static noise, which does not present in the baseline. Any insights?
  - It can be helpful to include discussion on both the pros and the cons of the proposed method.

**Summary Of The Paper:**

This paper proposes to use Gamma distribution noise to replace Gaussian distribution noise in the denoising diffusion probabilistic models (DDPM). It includes experimental results conducted on speech generation and image generation.


**Summary Of The Review:**

A sound idea with some strong experimental results. However, the presentation and the discussion can be improved.

---

> ### Author Response · Authors · 2021-11-18
> **Response to Reviewer YK5T**
>
> Thank you for the very helpful suggestions for corrections and improvements. All of these are now implemented.
>
> Q: Figure 2.
>
> A: The baseline samples came from the DDIM pretrained model that is provided in the official DDIM GitHub paper. The results in Ho et al., 2020 are obtained on an unpublished model that was trained on higher resolution CelebA images. Moreover, as noted in the caption, we use 100 iterations for generating the images. In Ho et al., 2020 1000 iterations were used.
>
> Q: Image experiments: since experiments on higher resolution were conducted (LSUN Church, 256x256), can you provide such image samples besides the lower resolution experiments?
>
> A: Thank you for pointing this out. At the request of the reviewer, these images are now added to the paper in Fig.3. We use the same noise instance for the DDIM and DDPM, therefore both models generate the same scene. As can be seen, our model has sharper and clearer images than the baseline DDIM model.
>
> Q: Speech experiment: The audio samples with few iterations sound significantly better than the baseline. However, all the audio samples from the proposed model with any number of iterations contain similar white-noise-like static noise, which does not present in the baseline. Any insights?
>
> A: You are correct. We have tried to investigate this very issue but we still don’t understand the link between the Gamma distribution and this type of noise. As part of our investigations, we employed a mixture distribution and were able to remove this noise.
>
> Q: It can be helpful to include discussion on both the pros and the cons of the proposed method.
>
> A: Pros - (i) Showing that the Diffusion models can also work with other types of noise distribution. (ii) Improve the results over the original methods like DDPM/DDIM. Cons- Gamma may not be the optimal distribution either. This discussion is now added to the paper in Sec. 5.

---

> > ### Comment · Reviewer_YK5T · 2021-11-27
> > **Reply**
> >
> > Thank authors for the responses.
> >
> > The significantly better results when a small number of iterations is used is valuable as it enables more efficient and lower latency models.
> >
> > The overall clarity of this paper on both the idea presentation and the experiments can be improved. The weakness of the proposed method, for example, the static noise in the speech experiment, worths more study and understandings.
> >
> > Therefore, I keep my ratings.

---

> > > ### Author Response · Authors · 2021-11-28
> > > **Dear reviewer**
> > >
> > >
> > > Thank you very much for your time and effort.
> > >
> > > We kindly want to emphasize that our proposed method improve the results for all of the vision datasets that we trained our model:
> > > Imagenet, lsun and celeba.
> > >
> > > Furthermore, our model improves the pseq and stoi scores, however, we have some noise in the generated samples. We are sure that this noise can be reduced with some classical signal processing techniques.
> > >
> > >
> > > Thank you.

---

### Official Review · Reviewer_C6QK · 2021-11-03

**Correctness:** 3
**Technical Novelty And Significance:** 3
**Empirical Novelty And Significance:** 3
**Recommendation:** 6
**Confidence:** 4

**Main Review:**

Although Gaussian noise has many attractive properties, it may be natural to ask whether other noise processes might be as good or better than Gaussian noise.  From that point of view this is an interesting paper and novel. The paper is reasonably well written, but I believe it can be improved in several ways.  The formulation can be written more clearly.  it would also be useful to expand on the motivation for the use of Gamma noise in the first place.  There is some discussion of this issue but it is relatively superficial. The experiments are good, but not extensive relative to other recent papers on generative models, including diffusion and score-based models.

1) Formulation:

The formulation of the variational lower bound in (22) involves KL terms like those in (26) for which there is no closed form expression.  The formulation in the paper instead provides an upper bound for the terms in (26), which is expressed in (27).  Once the terms in the lower bound are upper bounded, we no longer know whether the resulting objective remains a lower bound on the data log likelihood.  This appears problematic.

In the Gaussian case the KL terms of the objective, as shown by Ho et al, can be computed efficiently in closed form. In the Gamma case the computation of the objective is certainly more involved, which should be discussed explicitly in the paper.  It would be good to specifically address the computational cost of training DDGM vs DDPM and DDIM, for instance in Section 3.2.3.

One interesting property of the Gamma diffusion process is that repeated addition of independent Gamma distributed noise will become Gaussian over time due to the Central Limit Theorem. This implies that where the DDGM and DDPM deviate most is in the earliest steps of the forward diffusion process.  After 1000 steps, the noise distribution will have become a very close approximation to Gaussian noise.  As a consequence, it might seem somewhat surprising that DDGM and DDPM differ significantly in practice.  Is it possible to characterize over what range of time steps the Gaussian and Gamma cases are significantly different, and whether these differences are most significantly in the latest steps of the reverse process where fine-grained details are generated?


2) Experiments:  I have several very minor comments on the experiments.

a) It would be very useful to see results from DDGM on ImageNet (even ImageNet 64x64) as it is much more challenging and diverse than CelebA.  Ideally it would be great to see results in ImageNet 256x256 for which FID numbers are available for a wide range of techniques.

b)  In Table 3, how do FID numbers change as the number iterations increases to 500 or 1000?

c) In Figure 2, the two rows of face from different models appear very similar.  The paper says they are generated from the same noise sample?  But there is not a priori reason to believe that two independently trained model should produce similar results given the same latent noise sample. It would be good to explain this more clearly.

3) Minor issues of writing:

a) The paper states in multiple places that the sum of two Gaussian distributions is Gaussian and the sum of two Gamma distributions with the same scale parameter is also a Gamma distribution. This is not strictly true. The sum of two Gaussian random variables is Gaussian. But the sum of two Gaussian distributions is an unnormalized mixture model.  The same holds for Gamma random variables.  This is easy to fix of course.

b) The first paragraph of Section 2 states that the score function is the logarithm of the data density.  It should be the gradient with respect to the data of the log data density.

c) In Section 3.1 it might be useful state that the objective for the diffusion model is a variational bound on the model data log likelihood.  And in particular, it would be useful to remind readers that the KL terms in the objective have closed form expressions in the Gaussian case.

d) It might help readers to explicitly state that the noise in (9) is mean-zero.

e) Despite the result of Ho et al that DDPMs and denoising score matching are closely related, it does not appear that the most intuitive way to describe eqn (13) is in terms of Langevin dynamics.

f) The reverse process is specified in (14) as q(x_t-1 | x_t, x_0). While a non-Markovian model is fine, it might be useful to mention this to readers and cite the DDIM work.  It might be good to explicitly state the differences between DDPM and DDIM inn Section 3.3 as both are used later in the experiments.

g) People may find it more intuitive if the notation q(.) is reserved for the forward diffusion process, while the reverse diffusion model may be clearer if denoted by p_theta instead of q. For example, in the objective (Eqn 22) it is not clear at first glance where the parameterized model is. The definition of L0, for example, is given by q(x0 | x1), which is intractable, insted of the model p_theta(x0 | x1). The KL terms in (22), ie KL(q(x_{t-1} | x_t, x_0) || q(x_{t-1} | x_t, hat{x}_0)) hide the fact that hat{x}_0 is in fact an estimate obtained from x_t and the epsilon parameterization of the reverse process model. Improved notation in the paper would help the readability of the paper.

------

Post-Author Rebuttal:
Thank you for the clarification on the bound.  To avoid confusion it might be useful to bound the log likelihood rather than NLL and that way the bound is always a lower bound and my misunderstanding, mixing lower bounds in some places with upper bounds in others, can be avoided.  I also thank for the reviewers for the added quantitative results, etc in Table 3.
While most of my concerns have been addressed by the authors, I remain somewhat concerned that the difference between Gamma and Gaussian noise processes ceases to be substantial after several steps of the additive diffusion process.  The authors have agreed that the most significant difference will be at the beginning of the forward process (ie the end of the reverse, generative process), which makes sense.  In the end, I am not sure how much impact this work will have in the longer term -- will people begin choosing to use Gamma noise instead of Gaussian noise in practice?
While I can comfortably raise my score to 6 from 5, I still believe this is a borderline paper.

**Summary Of The Paper:**

This paper formulates a denoising diffusion probabilistic model, but with Gamma distributed noise instead of Gaussian noise. The claim is that the Gamma noise model shares many of the same useful properties as the Gaussian model (eg a variational bound on data log likelihood, and repeated application of Gamma noise remains Gamma distributed, etc).  And they show, empirically, that the Gamma model produces superior results on image generation (eg on the CelebA and LSUN Church datasets) and on speech generation (eg on the LJ dataset).


**Summary Of The Review:**

This paper describes a novel diffusion model based on Gamma noise instead of Gaussian noise.  The approach is shown to produce good empirical results compared to the Gaussian case.  But there are questions about the quality of the writing, the motivation for the use of Gamma diffusion, and the formulation, all of which the authors may want to address.  It would also be very useful to see more extensive experimental work on larger, more diverse data sets.

Post-Author Rebuttal:  While most of my concerns have been addressed by the authors, I remain somewhat concerned that the difference between Gamma and Gaussian noise processes ceases to be substantial after several steps of the additive diffusion process.  The authors have agreed that the most significant difference will be at the beginning of the forward process (ie the end of the reverse, generative process), which makes sense.  In the end, I am not sure how much impact this work will have in the longer term -- will people begin choosing to use Gamma noise instead of Gaussian noise in practice?  While I can comfortably raise my score to 6 from 5, I still believe this is a borderline paper.

---

> ### Author Response · Authors · 2021-11-18
> **Response to Reviewer C6QK**
>
> Thank you very much for the review. Your comments were much appreciated.
>
> **Formulation**
>
> Q: The formulation of the variational lower bound.
>
> A: The variational lower bound is an upper bound to the negative log likelihood (NLL). In (21) we upper bound the NLL, with the variational lower bound. In Eq(22-27) we continue to clarify the variational lower bound, and the direction of the inequality is always the same.
> No lower bounds were applied, and we follow the same pattern of bounds as in the DDPM paper. Specifically, the variational lower bound is a lower bound on the log likelihood and thus an upper bound of the negative log likelihood, which is used for the loss (see Eq 21). Thus our loss is truly an upper bound on the negative log likelihood.
>
> Q: The computational cost of training DDGM vs DDPM and DDIM.
>
> A: Comparing the computational cost of the DDPM vs DDGM, the only difference is the addition of one subtraction and deviation (the left term in Eq 28) which is O(1).
>
> Q: Is it possible to characterize ... in the latest steps of the reverse process where fine-grained details are generated?
>
> A: Following the review, we conducted a new experiment to show that the Gamma based denoising outperforms the baselines even with partial inference, i.e the improvement exists already during the first steps.
> In this experiment, we train the network on a Markov chain of size $T$ and perform all the processes until step $T/2$, from which we infer $x_0$ in one inference step. As can be seen, with the $T/2$ first iterations there is no significant difference between Gaussian and Gamma networks. However, for the remaining $T/2$ iterations the Gamma obtains a performance GAP over the Gaussian model. This suggests that our method is indeed improving the results mostly during the final part of the process, as predicted by the reviewer. These results are included in the revised version in Sec.4.
>
> |          | T=10 (first 5 steps/10 steps) | T=20 (first 10 steps/20 steps) | T=100 (first 50 steps/100 steps) |
> |----------|-------------------------------|--------------------------------|----------------------------------|
> | Gaussian | 54.32/17.33                   | 43.35/13.73                    | 36.68/6.53                       |
> | Gamma    | 55.40/11.64                   | 42.43/6.83                     | 35.20/3.17                       |
>
>
> **Experiments**
>
> Q: Results from DDGM on ImageNet...
>
> A: Following the review, we trained the DDGM and DDPM on ImageNet 64x64 (due to the time constraint). The results are in the table below. Evidently, DDGM outperforms DDIM. We add this table as Tab.4 in the revised manuscript.
>
> | Imagenet 64x64 (FID) | T=10  | T=20  | T=50  | T=100 | T=1000 |
> |----------------------|-------|-------|-------|-------|--------|
> | Gaussian (DDIM)      | 42.88 | 35.40 | 31.98 | 30.74 | 28.81  |
> | Gamma (DDIM)         | **42.17** | **31.84** | **28.75** | **27.02** | **24.22**  |
>
>
> Q: In Table 3, how do FID numbers change ... 1000?
>
> A: Following the review, the following would be added to the table.
>
> |                            | T=1000                                                    |
> |----------------------------|-------------------------------------------------------------|
> | DDPM                       | 7.89                                                        |
> | DDGM (with DDPM inference) | 6.91                                                        |
> | DDIM                       | X (the DDIM paper does not report this for 1000 iterations) |
> | DDGM (with DDIM inference) | 7.34                                                        |
>
>
> Q: In Figure 2, It would be good to explain this more clearly.
>
> A: This is indeed a very interesting phenomenon of the DDIM sampling process. With independently trained neural networks, it generates images that are often very close. This doesn’t happen when using DDPM sampling since each step introduces new random noise. In the case of DDIM, the noise is sampled only once and is reused at each step (see Eq 12 in Song and Ermon DDIM paper.).
>
> **Minor issues of writing**
>
> a) Thank you for this comment, in the revised version, this is corrected in Sec.1 (Introduction) and in Sec. 3.2.1 (The Gamma Model).
>
> b) Thank you for this comment, in the revised version, this is corrected in Sec.2.
>
> c) Thank you for this comment, in the revised version (Sec. 3.1 & Sec. 3.2.3) we note that the Gaussian case has a closed form.
>
> d) Thank you for this comment, a note about the zero mean has been added to the revision in Sec.3.2.1
>
> e) Thank you for this comment, the Langevin dynamics reference in Eq13 is not included in the revision (Sec. 3.2.1.).
>
> f) Thank you for this comment, the revision elaborates on the DDIM method and the difference from DDPM in Sec.3.1.
>
> g) Thank you for this comment, in the revised manuscript, we change the reverse process to be p_theta and the forward to be q.

---

### Official Review · Reviewer_nekw · 2021-11-10

**Correctness:** 3
**Technical Novelty And Significance:** 2
**Empirical Novelty And Significance:** 2
**Recommendation:** 5
**Confidence:** 3

**Main Review:**


## Strengths
* Interesting extension to diffusion models, a new, relatively underexplored class of deep generative model.
* Detailed derivations that are easy to follow.

## Main concerns and questions
* Lack of motivation:  I think it's great that the paper shows sample quality improvements on both image and audio dataset.  However, I have a few concerns.
  * Towards higher iteration count, the gap between DDPM and DDGM seems to decrease (Tables 1, 2).  I'm curious if DDPM and DDGM eventually converge to a similar performance number, if we don't limit the iteration count.  If they do converge or the improvement becomes negligible, then DDGM feels more like a method to improve sampling efficiency (in the same vein as e.g. DDIM), rather than a way to actually improve the expressivity of the model.
  * In Table 2, is there any explanation for why DDPM eventually outperforms DDGM at 1000 step count?
  * I'm not quite sure if I understand Figure 1 and the explanation given in the Introduction.  In what way is the ability to fit the histogram of marginal pixel values better (presumably aggregated across all dimensions) related to the overall expressivity of the resulting DDGM?
* Experimental rigor: As far as I understand, the original DDPM model was trained and evaluated on CIFAR10 and LSUN bedroom.  Presumably, the hyperparameters are tuned for those datasets.  So I'm a bit concerned that, while the model itself is based on the DDIM implementation, the hyperparameters are taken from the DDPM paper -- especially for LSUN Church, where there is very small gap between DDIM/DDPM and DDGM for several iteration counts.
* (Minor comment) The current version of the manuscript I believe has an unnecessary amount of mathematical detail in the main text.  It'd improve the overall readability of the paper if some of the detailed derivations and algebraic manipulations are moved to the supplementary material, since they are not important part of the overall story.

--------------------------------------------------------
POST REBUTTAL: Based on the authors' response, I raised the score to 5.

**Summary Of The Paper:**

## Summary
This paper explores the use of a non-Gaussian diffusion process for Diffusion Probabilistic Models.  Unlike the original work by Ho et al., the authors replace the diffusion process with a Markov chain with transition kernel defined by a Gamma distribution.  They show that the similar (and necessary) properties of Gaussian distribution that enable training DPM in practice also hold true for Gamma distribution.  The main motivation why Gamma distribution is used seems to be that Gamma distribution is more expressive than Gaussian due to having an extra parameter.  The authors experimentally verify the performance gains on a few datasets.


**Summary Of The Review:**

## Overall thoughts
The paper is well-written, and I had no difficulty understanding it for the most part.  I'm a bit concerned by the experimental methodology and (more importantly) the motivation for using Gamma distribution.  If the authors can better address these issues, I am willing to adjust my review accordingly.

---

> ### Author Response · Authors · 2021-11-18
> **Response to Reviewer nekw**
>
> We thank the reviewer for the extremely helpful feedback.
>
> Q: ..I'm curious if DDPM and DDGM eventually converge to a similar performance number, if we don't limit the iteration count.
>
> A: Referring to Tab.2 in the submitted manuscript, one can observe that in the LSUN dataset, for a larger number of iterations (e.g. 1000) our model improves the results compared to DDIM. Due to the diminishing returns effect, as the number of iterations increases the improvement becomes smaller. Therefore, the most beneficial one can obtain from our method is in the regime of lower iterations. In the revised manuscript, we also add results for LSUN church with 1000 iterations, and our DDGM model improves the baseline results.
>
> At the request of reviewer C6QK, we have conducted experiments on ImageNet 64x64. As the table below shows, our method improves by the same FID gap across multiple numbers of iterations. We add this table as Tab.4 in the revised manuscript.
>
> | Imagenet 64x64 (FID) | T=10  | T=20  | T=50  | T=100 | T=1000 |
> |----------------------|-------|-------|-------|-------|--------|
> | Gaussian (DDIM)      | 42.88 | 35.40 | 31.98 | 30.74 | 28.81  |
> | Gamma (DDIM)         | **42.17** | **31.84** | **28.75** | **27.02** | **24.22**  |
>
>
>
> Q: In Table 2, is there any explanation for why DDPM eventually outperforms DDGM at 1000 step count?
>
> A: The hyperparameters used for DDGM were the ones optimized on DDPM and we believe that the results for DDGM can be improved. Note that for DDIM,  which is less sensitive to hyperparameters, our method outperforms the baseline.
>
> Q: Figure 1?
>
> A: Figure 1 depicts the histogram of $ \frac{\sqrt{\bar \alpha_t}x_0 - x_t}{\sqrt{1 - |\bar \alpha_t|}} $ after (k) generation steps with a gaussian DDIM. The model assumes that this quantity is distributed in accordance with a  Gaussian distribution. However, evidently, it is not. In other words, since the model is trained only to predict Gaussian noise it is clear that the training task is not adequately chosen with regard to the inference task that it will have to perform. We are therefore motivated to employ a model that predicts a more suitable noise distribution. We add this explanation in the revised version.
>
> Q: Hyperparameters for DDIM.
>
> A: DDPM and DDIM are two sampling methods based on the same model. Therefore, the hyperparameters are the same for both. In order to show that our performance improvement is not based on a hyperparameter search, we decided to use the same parameter as previous literature.
> Regarding the relatively small improvement: Indeed, for some iteration counts the gap is not extremely large. However, the FID score is computed with 50,000 samples and the differences are statistically significant across all iteration counts.
>
> Q: Amount of mathematical detail in the main text.
>
> A: Following the review, in the revised version we move the “the reverse process for DDGM” and the “upper bound for $L_{t-1}$” sections to the supplementary material as two new lemmas.

---

### Decision · Program_Chairs · 2022-01-20

**Decision:**

Reject

**Comment:**

This paper explores replacing the Gaussian noise typically used in diffusion-based generative models with noise from other distributions, specifically the Gamma distribution. The effect of this change is studied empirically for both image and speech generation.

Reviewers welcomed the exploration of the design space of diffusion models, and several reviewers consider the study of alternative noise distributions in particular an important contribution. They also raised several issues with precision and clarity (several mistakes in the manuscript were pointed out), the quality of the experiments, and, especially, a lack of convincing motivation for this exploration / sufficient demonstration of its impact.

While the authors have made a significant effort to address the reviewers' comments and suggestions, which includes running additional experiments, all reviewers have nevertheless chosen borderline ratings, with half erring on the side of rejection, and the other half tentatively recommending acceptance.

I am inclined to agree that, as it stands, the benefit of the proposed change of noise distribution is not convincingly shown to outweigh the additional complexity this introduces, so I am also recommending rejection.